# Low-Shrinkage Resin Matrices in Restorative Dentistry-Narrative Review

**DOI:** 10.3390/ma15082951

**Published:** 2022-04-18

**Authors:** Ebtehal G. Albeshir, Rashed Alsahafi, Reem Albluwi, Abdulrahman A. Balhaddad, Heba Mitwalli, Thomas W. Oates, Gary D. Hack, Jirun Sun, Michael D. Weir, Hockin H. K. Xu

**Affiliations:** 1Program in Dental Biomedical Sciences, University of Maryland School of Dentistry, Baltimore, MD 21201, USA; ealbeshir@umaryland.edu (E.G.A.); rashed.alsahafi@umaryland.edu (R.A.); 2Department of Restorative Dentistry, King Abdul-Aziz Medical City, Ministiry of National Guard—Health Affairs, Riyadh 11426, Saudi Arabia; albluwire@ngha.med.sa; 3King Saud Bin Abdulaziz University for Health Sciences, Riyadh 11426, Saudi Arabia; 4King Abdullah International Medical Research Center, Ministiry of National Guard—Health Affairs, Riyadh 11426, Saudi Arabia; 5Department of Restorative Dental Sciences, College of Dentistry, Umm Al-Qura University, Makkah 24381, Saudi Arabia; 6Department of Restorative Dental Sciences, College of Dentistry, Imam Abdulrahman bin Faisal University, Dammam 31441, Saudi Arabia; aabalhadd@iau.edu.sa; 7Department of Restorative Dental Science, College of Dentistry, King Saud University, Riyadh 11451, Saudi Arabia; hebamitwalli@gmail.com; 8Department of Advanced Oral Sciences and Therapeutics, School of Dentistry, University of Maryland, Baltimore, MD 21201, USA; toates@umaryland.edu (T.W.O.); ghack@umaryland.edu (G.D.H.); 9The Forsyth Institute, A Harvard School of Dental Medicine Affiliate, 245 First Street, Cambridge, MA 02142, USA; 10Center for Stem Cell Biology & Regenerative Medicine, University of Maryland School of Medicine, Baltimore, MD 21201, USA; 11Marlene and Stewart Greenebaum Cancer Center, University of Maryland School of Medicine, Baltimore, MD 21201, USA

**Keywords:** dental material, resin composite, low polymerization shrinkage, oral biofilm, secondary caries, microleakage, hydrolytically stable resins, nanocomposite

## Abstract

Dimethacrylate-based resin composites restorations have become widely-used intraoral materials in daily dental practice. The increasing use of composites has greatly enhanced modern preventive and conservative dentistry. They have many superior features, especially esthetic properties, bondability, and elimination of mercury and galvanic currents. However, polymeric materials are highly susceptible to polymerization shrinkage and stresses that lead to microleakage, biofilm formation, secondary caries, and restoration loss. Several techniques have been investigated to minimize the side effects of these shrinkage stresses. The primary approach is through fabrications and modification of the resin matrices. Therefore, this review article focuses on the methods for testing the shrinkage, as well as formulations of resinous matrices available to reduce polymerization shrinkage and its associated stress. Furthermore, this article reviews recent cutting-edge developments on bioactive low-shrinkage-stress nanocomposites to effectively inhibit the growth and activities of cariogenic pathogens and enhance the remineralization process.

## 1. Introduction

The application of biomaterials science is unique in dentistry because of the complexity of the oral cavity. The oral environment is considered the most challenging for material in the body, in which high mechanical loading, bacteria, changing pH, and a warm, fluid environment [1]. Understanding the physical, chemical, biological, and biocompatible properties of restorative material is very important in determining their application [2]. Although tooth form and appearance are easily recognized, the proper function of the teeth and their supporting tissues have a great impact on life quality, including eating, swallowing, speaking, and breathing [3]. Thus, adequate selection of restorative materials is essential to achieve the desired purposes. Restorative dental materials are generally categorized into the following classes: metals, ceramics, polymers, and resin composites [1,2].

Resin-based composites have been introduced into conservative dentistry to minimize the drawbacks of acrylic resins and silicate cement, the only aesthetic materials previously available [4]. Since their development in the late 1950s [4], resin composites have become transformational and widely used intraoral materials in restorative dentistry [2,5]. Recently, patients are more attracted to tooth-colored restorations, so manufacturers have developed resin materials that allow dentists to place artistic restorations matching the natural form and color of the teeth by optimistic handling and optical properties [6,7]. Besides their excellent esthetic properties, resin composites have low thermal conductivity, radiopacity, no mercury, and no galvanic currents [7,8]. In addition, they can directly adhere to tooth structure without the need for excessive removal of sound hard tissues, eliminating the “extension for prevention” technique [8]. This bondability—achieved by applying an adhesive—seals the margins and reinforces the remaining tooth structures. The bonding succeeded by using acids that etched the dental tissues to form numerous micromechanical interactions between the restoration and enamel and dentin [8]. The variety of dental tissues also plays a significant role in the adhesion process, for example, enamel vs. dentin tissues, permanent vs. deciduous teeth, or vestibular vs. lingual dental tissues [9]. Because of the material’s bonding, it has increased application in modern preventive and conservative dentistry. As a result, resin composites can be used for different purposes such as anterior and posterior teeth injured or diseased by the caries process, occlusion adjustments, orthodontic bonding brackets, and aesthetic tooth modifications [10,11,12].

Composites are a three-dimensional combination of four major chemically different components, Figure 1. The matrix is the continuous phase that allows the other component to blend in. Typically, it consists of high viscosity polymers and diluent monomers, with small percentages of initiators and inhibitors to control the polymerization reactions. Most of the commercially available composites are composed of dimethacrylate-based matrices, especially 2,2-bis-[4-(2-hydroxy-3-methacryloyloxy propoxy) phenyl] propane (Bis-GMA) and urethane dimethacrylate (UDMA) or a combination of both [13]. Low molecular-weight monomers, such as triethylene glycol dimethacrylate (TEGDMA) or Ethoxylated (six-mole ethylene oxide) Bisphenol A Dimethacrylate (Bis-EMA6), are added by the manufacturers to improve clinical consistency and overcome the high viscosity of Bis-GMA or UDMA [13]. Polymer terminals are combined through additional polymerization reactions. The reaction is initiated by unpaired free radicals generated by the initiators [14]. The organic matrix is usually considered the problem phase because of the high shrinkage rate after the polymerization process, where most of the restorations’ drawbacks have occurred.

The dispersed inorganic filler contains radiopaque particles that may consist of finely ground glass, microfine silica, ceramics, or nanoparticles [15]. The inorganic filler particles resolve most of the matrix’s drawbacks. They decrease the polymerization shrinkage and improve the strength and the final physical properties of the polymerized composites. The coupling agent or organosilane is applied to the surface-treated fillers to create a chemical bond between the organic and the inorganic phases to form a cohesive manipulable composite paste [15].

Although resin composite has potent positive features over the other filling materials, they have their own negative aspects. First, dental resin composites lack antibacterial or remineralization properties, which play an essential role in caries formation. The main oral pathogens include anaerobic *Streptococcus mutants* and *Lactobacilli*. The bacterial by-products are accumulated via specific binding proteins on the interface between the restorative materials and dental tissues. Then, a thicker biofilm will reduce the pH of the area, leading to demineralization, secondary dental caries, and periodontal diseases [16,17]. Second, they have relatively less-than-ideal wear resistance, especially in bruxism patients, and cuspal replacement in high-bearing areas [3,8]. Third, they are subjected to swelling because of water sorption that leads to hydrolytic breakdown and debonding. Fourth, they have lower fracture toughness and lower modulus of elasticity compared to amalgam restorations [14]. Lastly, their coefficient of thermal expansion and contractions are significantly different than the tooth structure when subjected to high or low temperatures. As a result of their inferior physical properties, resin composites may develop microcracks or bulk fractures that negatively affect their longevity [1,14].

Dimethacrylate polymer-based materials also have drawbacks related to polymerization shrinkage and associated stresses [18]. Typically, composites used in dental procedures exhibit volumetric shrinkage toward their center, ranging from less than 1% to 6%, depending on the formulation and curing conditions [19]. The polymerized resin is highly cross-linked because of difunctional carbon double bonds. Usually, 35% to 80% of the double bonds polymerize to transform the monomers into polymer [20]. As a result of this, polymerization stress should start to develop within the polymer throughout the different stages of the transformation, significantly if the material is constrained by the adhesion to the cavity walls of the restoration [18].

During photopolymerization of a resin composite, it is transformed from a fluid viscous paste to a glassy hard material [18]. Initially, the resin has the freedom of motion flow and partially relieves the stresses. After light-cure exposure, the monomers become reactive to form a polymer chain, and the distance between the monomers is reduced to form the cross-linked network [19]. Hence, there is a gradual increase in the viscosity and loss of the polymer fluidity (gel-point); the elasticity may remain low enough to lead only to low stresses. Next, the composite will lose the flowing ability and change to a rigid glassy mass (vitrification point) [18]. Moreover, the modulus of elasticity and polymerization shrinkage becomes high, and stress relaxation becomes slow. Therefore, any restraints on the polymerization shrinkage will generate residual shrinkage stresses. As a result, high-stress magnitude begins to emerge significantly faster when the curing reaction proceeds from pre-gel to vitrification [18,19,20].

Furthermore, the bonding between the organic matrix and the glass particles and differences in their coefficient of thermal expansion during the polymerization reaction can increase the magnitude of the stresses. Additionally, the surrounding conditions around the restoration, such as bonding integrity, configuration factor, and deformation of the adjacent structures, are essential for straining the interfacial bond between the composite and the tooth [21,22]. The physical outcomes associated with shrinkage stresses are several and related to a reduction of the restoration longevity [14,16,20], Figure 2.

Reducing the polymerization shrinkage can be addressed by modifying the material’s formulation, such as the resin matrix composition or the fillers. In addition, polymerization shrinkage stresses can be reduced by applying techniques such as incremental layering, a stress-absorbing base, or a liner [6,14]. Furthermore, modifying light-activation protocols, such as soft-start and pulse delay, have been advocated to reduce shrinkage stress [20].

The hypothesis is that starting the polymerization with low intensity produces a reduced amount of free radicals with a slower polymerization reaction, delaying the vitrification point of composite [20,23]. Thus, it allows more relief of shrinkage stress by prolonging the period that composite can flow [24]. Based on this concept, many light-curing units offer alternative regimens to emit light on pulsatile, ramp, or soft-start modes [23,24,25]. However, many studies showed that the set composite using modified light-activation protocols might be more prone to degradation, lower elastic modulus, and increase the risk of failure under occlusal loading because of lower strength properties [25,26].

Multiple evolutions improved resin composites’ esthetics, handling, and physical properties. However, the manifestation of the polymerization shrinkage and the associated stresses are the major problems facing manufacturers and clinicians.

## 2. Materials and Methods

This review paper focuses on polymerization shrinkage and stresses in dental composite restorations. It sheds light on the different approaches to physical testing of the stresses and measuring the shrinkage. Furthermore, this article discusses the formulations of resinous matrices available to reduce shrinkage and the associated stresses. In addition, this review updates the reader with cutting-edge research on bioactive low-shrinkage-stress nanocomposites.

For Section 6.1, a research library supported the databases search for subject terms, keywords, and text words, that were related to studies on low-shrinkage resin matrices in restorative dentistry. The related articles were searched using Medline (OVID) and EMBASE databases. The investigation method conducted for MEDLINE was followed for EMBASE and appropriately revised to account for vocabulary differences. Searched terms were related to the low-shrinkage resin matrices used in dentistry and involved, but were not limited to, antibacterial or antibiofilm, remineralization, resin or composite or nanocomposite, and low shrinkage stresses. The searches were limited to peer-reviewed journals. Searches were also limited to English-language articles from 2000 to 2022. At the end of the search, only six qualified studies were found.

## 3. Approaches for Testing Resin Composite Shrinkage

Many approaches aim to measure the volumetric changes during the different stages of the polymerization reaction. Efforts were also made to measure the generated stresses. These methods help researchers and clinicians better understand, predict, and manage the polymerization shrinkage of dental composites.

### 3.1. Coordinate Measuring Systems

Coordinate measuring systems are easy and quick assessment methods of polymerization shrinkage using automated methods. The main objective of this coordinate system is to find the Cartesian coordinates of points on a plane, a set of numerical coordinates [27]. A Coordinate measuring machine (CMM) comprises one fixed-based, three movable parts, and four rigid parts. The most commonly used type is CMM with a movable bridge and fixed granite table. Each of the three movable parts can be moved by an operator along the X, Y, and Z axes.

Then, a specific point on the surface is detected by a probe that has a sensor to locate the planes to determine the volume of the given object. Polymerization shrinkage is computed according to the volume differences of the object at two different times [28].

### 3.2. Optical Coherence Tomography

Optical coherence tomography (OCT) is an imaging method based on a Michelson interferometer. OCT uses low-coherence light to take dimensional images from optical scattering media. This method consists of three parts: a base unit includes the superluminescent diode light source, scanning probe, and computer [29]. For this method, first, an empty mold is scanned to capture the accurate mold thickness. Then a second scan is made after the insertion of the composite to calculate the amount of unpolymerized composite. Finally, fifteen minutes after light curing, the last scan is made. Linear polymerization shrinkage is then analyzed according to the following formula:(1)$$\mathrm{Liner}\;\mathrm{shrinkage}=\frac{RC_{0min}-RC_{15min}}{RC_{0min}}\times 100\%$$
where RC_0min_ is the mean composite thickness in the uncured state and RC_15min_ is the mean composite thickness of the cured state [28].

### 3.3. Archimedes’ Principle

The buoyancy of material in fluid or Archimedes’ principle is a well-known experiment technique that can use to calculate volumetric changes [30]. This principle states that at rest, any immersed body (partially or completely) in a gas or liquid is acted upon by an upward buoyant force proportional to the dispersed fluid weight. In this method, density measurements determined volumetric shrinkage according to Archimedes’ principle by measuring the specimen’s weight several times in two different environments of known density. According to the below equation, the density of the specimen is calculated [31]:(2)$$\rho =\frac{m_{water}}{m_{air}-m_{water}}\left(\rho_{water}-\rho_{air}\right)+\rho_{air}$$
where ρ is the material density, m_air_ is the specimen weight in grams in air, m_water_ is the specimen weight in grams in water, ρ_air_ is the air density (0.0012 g/cm), and ρ_water_ is the water density at the precisely measured temperature according to the distilled water density.

### 3.4. Strain Gauge

This method is susceptible to changes in the linear dimensional [29]. In the strain gauges method, the changes in linear dimensional occurring in a substrate are moved to a gauge bonded to it and calculated. This change in linear dimensional is only transferred when the substrate has a measurable post-gel modulus to produce stress on the gauge. Thus, this method is appropriate for measuring post-gel shrinkages of composites [32].

### 3.5. Dilatometer Method

Dilatometry is one of the most commonly used methods for measuring the volumetric change of polymerized materials [31]. In this technique, the composite sample is surrounded by a non-reacting liquid, such as mercury, during the curing period. Monitoring the alteration in the volume of a mercury column allows an operator to measure the volumetric changes related to the sample’s polymerization. Accordingly, the shrinkage is recorded during the curing period from the pre-polymerization to post-polymerization phases [32]. Thus, the degree of total shrinkage is recorded.

### 3.6. Modified Bonded Disk Method

An uncured composite past is placed onto a rigid glass plate in this method. A thin microscope coverslip is placed on the composite sample, and a brass ring is positioned around the composite specimen to support the glass coverslip. The brass ring’s internal diameter must exceed the diameter of the sample. By measuring the thin glass coverslip’s deflection recorded by a linear vertical displacement transducer, shrinkage is indirectly measurable [32].

### 3.7. Thermomechanical Analyzer Method

Thermomechanical Analysis (TMA) provides valuable information on the polymerization shrinkage of dental composite. In this technique, dimensional changes of the sample under non-oscillating stress are monitored against time or temperature. For dental composite polymerization shrinkage measurement, a borosilicate glass cylinder is bonded to a glass plate and placed on a TMA quartz sample stage. The glass cylinder’s axial wall is coated with a thin layer of a release liquid to prevent adhesion between the axial surfaces and the composite. A quartz dilatometry probe is positioned on the composite specimen in the glass cylinder. Then, a curing light is used to start the polymerization of the composite. Polymerization shrinkage is calculated by dividing the displacement by the initial specimen height [32,33].

### 3.8. X-ray Microcomputed Tomography

X-ray microcomputed tomography (μCT) can acquire three-dimensional (3D) structures of the internal content of small objects with high spatial resolution [34]. Biomedical research has been widely accepted for examining bone and tooth structures, visualizing structural features in tissue engineering scaffolds, and assessing the mineral concentration of teeth [35]. Recently, μCT has been used to evaluate the junction of the tooth-adhesive and the marginal adaptation, which correlated with shrinkage strain of the resin composite restorations [36,37]. The previous study used μCt to examine the dimensional changes of dental resin composites before and after polymerization [35]. The acquired results agreed with the degree of shrinkage achieved by density measurements. The μCT methods give matching accuracy for different physical states and shapes. Furthermore, the μCT results are not affected by air bubbles because they are not included in calculating the dimension of the resin restoration [35,37].

### 3.9. Video Imaging Methods

Accurate volumetric shrinkage can be measured using video imaging methods like AcuVol and Drop Shape Analysis System. This technique allows comparing the volumes of resin restoration before and after polymerization using a video imaging method.

One of the most significant advantages of this technique is its ease of usage and capability to follow volumetric shrinkage during the entire curing time. This method has been exhibited to yield results similar to those obtained using mercury dilatometry [38].

### 3.10. Cantilever Beam-Based Tensometer

Cantilever beam-based tensometers measure the polymerization shrinkage stress, curing kinetics, and degree of conversion, such that a better understanding and control of the curing process can be achieved. The setup consisted of the tensometer with a built-in high-speed NIR spectrometer, which permits simultaneous monitoring of the real-time double-bond conversion in transmission. In this technique, a disk-shaped (2.5 mm in diameter and 2 mm in height) uncured specimen will be placed between two flat methacrylate-silane treated quartz rods. The upper rod was clamped to the cantilever beam, and the lower one was fixed to the base. The specimen will be photoactivated with a curing light with specific intensity at the top end of the lower rod where the specimen is attached. Polymerization shrinkage occurred, and the resulting axial shrinkage stress caused a deflection in the beam. This deflection was recorded by a displacement sensor at the free end of the beam and used to deduce the axial stress based on a beam formula:(3)$$\sigma =F/A=6\delta EI/\pi r^{2}a^{2}\left(3\iota -a\right)$$
where s is the polymerization shrinkage stress, F is the force exerted by the sample shrinkage, A is the cross-sectional area, r is the radius of the sample, d is the beam deflection at the free end, E is Young’s modulus, ι is the moment of inertia of the beam, and l and a are the length of the beam and the distance between the sample position and the clamped edge of the beam, respectively [39,40].

## 4. Resin Formulations Attempt to Reduce Shrinkages and Their Stress

Several investigations were attempted to reduce the polymerization shrinkage via the manipulation of the filler size, load, and configuration [15]. However, the amount of polymerization shrinkage reduction was limited, which has led to efforts to explore other strategies via the use of different resin matrix systems.

### 4.1. Silorane-Based Resin Composite

The silorane matrix polymerization in silorane-based resin composites (SBRC) is achieved by the ring-opening resulting in lower volumetric changes compared to the methacrylate-based resin composite (MBRC) [41]. The siloxane core within the silorane molecule contains four oxirane rings. When the oxirane rings are subjected to polymerization, they open and bond to other monomers. It has been suggested that the opening of the oxirane rings results in a volumetric expansion, which may compensate for the shrinkage resulting from polymerization [42]. The polymerization shrinkage in SBRC was reported to be less than 1%, which can subsequently reduce the polymerization stress, gap formation, cusp deflection, and enhance the marginal seal and adaptation [42,43]. One of the main advantages of SBRC is related to its gloss and hydrophobicity. Less water absorption due to hydrophobicity means that the staining of the restoration and the adhesion of microorganisms is less problematic in SBRC [44].

Several reports were conducted to compare SBRC and MBRC. One study compared SBRC to four MBRC products [43]. Using three different light-curing units, the least polymerization shrinkage was observed in SBRC (7.58–8.91 μm), while the MBRC products reported polymerization shrinkage ranging between 9.69 to 14.17 μm [43]. Another study also found that the volumetric shrinkage of SBRC (0.88 ± 0.04) is significantly less than MBRC, which ranged between 1.75 to 1.97% [45]. The same results were reflected in another study where the volumetric shrinkage of SBRC was 1.76 ± 0.03, compared to 2.78 ± 0.08 observed in the MBRC group [46]. A recent meta-analysis including nine studies revealed that SBRC was associated with significantly reduced shrinkage upon polymerization compared to other resin composite systems [47]. Nevertheless, other studies reported that the polymerization kinetics of SBRC is comparable to MBRC [45,48]. The same controversy was observed when evaluating the mechanical properties as some investigations showed that the mechanical properties of SBRC were less compared to MBRC [49,50]. One study reported that the flexural strength values of SBRC were comparable to MBRC, but the compressive strength values were the lowest [43]. The same observation was reported in another study as the flexural strength and fracture toughness values of SBRC were high, but the compressive strength was relatively low [51].

In regard to microleakage, it is logical to speculate that composites with less shrinkage should have less microleakage, which was reported by one study that found SBRC to induce less microleakage after thermocycling and mechanical loading compared to MBRC [52,53].

However, another investigation found no significant difference in microleakage score between SBRC and MBRC [45]. Less cusp deflection was reported with the use of SBRC [50], but the opposite was reported as well [54].

Camphorquinone is used as a photoinitiator in SBRC; therefore, the same light-curing unit can be used for both SBRC and MBRC as the wavelength is similar. However, it is important to mention that iodonium salt and ethyl dimethylaminobenzoate are incorporated in SBRC. As a result, another bonding system that is specific to SBRC must be used [41]. Several studies were conducted to evaluate the bonding strength of SBRC. While some studies found no significant difference between SBRC and MBRC [55,56], others observed that MBRC is associated with higher bonding strength [57,58]. It is critical to mention that the bonding strength should be higher than the shrinkage stress to retain the restoration. In a cavity with a high C-factor, the amount of shrinkage stress is expected to be high, which may compromise the bonding strength of such restorations [59]. One investigation reported that SBRC micro tensile bonding strength was 25% less than MBRC when the composites were bonded to a flat surface [59]. However, when the two systems were bonded to Class I cavity models (high C-factor), no significant difference was observed [60]. The explanation of such observation is that the bonding strength of MBRC was reduced due to the polymerization shrinkage. The effect of the shrinkage was less pronounced in SBRC as the amount of the shrinkage is less. Based on these observations, both SBRC and MBRC may have comparable bonding strength when bonded to clinically relevant cavity models [60].

Several clinical studies were conducted to compare the clinical performance between SBRC and MBRC. In one study, 29 patients received 29 pairs of restorations placed in class II cavities. After three years, no significant difference was found between the two materials regarding the anatomic form, color match, roughness, marginal discoloration, secondary caries, and sensitivity. However, in regard to the marginal adaptation, MBRC restorations were significantly better as 58.62% of the SBRC restorations were reported with explorer catch [61]. The same observation was stated in another article where MBRC demonstrated better marginal adaptation than SBRC [62]. A five-year clinical study compared SRBC and MBRC placed in class II cavities and found no difference concerning approximal contact, anatomic form, fractures, discoloration, secondary caries, and sensitivity [63]. In general, most of the clinical studies reported similar clinical outcomes between SBRC and MBRC [64,65]. Based on that, using SBRC may not provide any advantage regarding the clinical longevity of resin composite restorations.

### 4.2. Nanogels and Functionalized-Nanogels

The incorporation of nanogel prepolymer overcomes the limitation found in previous reports with the use of polymer with large molecules. Incorporating large molecules as fillers may result in water sorption, thermal expansion, and reduced modulus of elasticity [15,20]. Opposingly, the use of nanogel in its nanoscopic size allows a greater number of fillers to be incorporated. Replacing the monomer with prepolymerized nanogel minimizes the reactive group concentrations and lowered the volume alteration during the polymerization reaction [66]. In 2011, the use of nanogel particles as a strategy to lower the polymerization shrinkage and stress was proposed [66]. Isobornyl methacrylate and urethane dimethacrylate were mixed at the ratio of 70:30 to synthesize the nanogels. 2-Mercaptoethanol was incorporated as a chain transfer and to allow further functionalization with the existing methacrylate groups. The methacrylate-functionalized reactive nanogels were obtained via a specific precipitation approach and mixing the nanogels with 2-isocyanoethyl methacrylate. Reactive and non-reactive nanogels were mixed with triethylene glycol dimethacrylate (TEGDMA) at 5, 10, 20, 30, and 40 wt%. Then, conventional resin composite formulations were designed by incorporating barium glass filler at 70 wt% to TEGDMA containing the nanogels [64]. The kinetic profile of the TEGDMA control was similar to that nanogel-modified TEGDMA despite the type of nanogels (reactive or non-reactive).

Both reactive and non-reactive nanogels significantly reduced the polymerization shrinkage of TEGDMA by 37–43%. In the composite formulation, 40 wt.% nanogels lowered the polymerization shrinkage by 24%. No significant difference was observed concerning the flexural strength and elastic modulus when the reactive nanogels were incorporated into TEGDMA resin or the composite formulation. However, increasing the load of the non-reactive nanogels was associated with reduced flexural strength values [67]. In another report, the nanogel prepolymer was modified to incorporate polysiloxane and improve the surface characteristics of the synthesized polymer [68]. The polysiloxane-modified nanogel was incorporated into TEGDMA at 10, 20, 30, and 40 wt.%. Similar kinetics behavior was reported in both TEGDMA control and TEGDMA with polysiloxane-modified nanogel. Incorporating the nanogel into TEGDMA significantly reduced the polymerization stress and shrinkage. Incorporating the polysiloxane resulted in a heterogenous monomer system with improved thermal stability [68].

Plasticizers as low molecular weight monomers are commonly used to enhance polymer flexibility and manipulate the glass transition temperature. Plasticizer migration presents a real challenge in polymer design as it may affect the mechanical properties. Using aromatic thiols was found beneficial to resist plasticizer migration and improve the stability of the polymer [69]. Thiol-functionalized nanogels were designed to minimize the plasticizer migration, lower the transition, and improve the elongation properties without affecting the thermal stability. Additionally, a 52% reduction in the polymerization stress was observed [70]. Another report functionalized a coupling agent into nanogels. γ-methacryloxypropyltrimethoxysilane (MPS) as a coupling agent is important to link the resin matrix and the loaded fillers for mechanical reinforcement and minimize the degradation process. However, it seems that the MPS functional group is not stable, resulting in stress development during the polymerization reaction [71,72]. As a result, several investigations attempted the use of alternative coupling agents such as hyperbranched oligomers and thiourethane-modified silane to lower the shrinkage stress. A recent investigation proposed the use of amine functional silanes as well as isocyanate-methacrylate nanogels to reduce the polymerization stress [73].

### 4.3. Dimethacrylate-Derivative of Dimer Acid

The use of dimer acid-based monomer to reduce the polymerization shrinkage is based on the high molecular weight with the lowered concentration of the initial double-bond. This monomer was reported with reduced polymerization shrinkage and a high degree of conversion [74]. A commercial resin composite based on dimethacrylate-derivative of dimer acid was formulated (N’Durance, Septodont company. Saint-Maur-des-Fosses, France) by incorporating it with BisGMA, urethane dimethacrylate (UDMA), and dicarbamate with a filler load of 80% by weight [73]. This commercially available low shrinkage composite was reported with higher polymerization stress and shrinkage compared to other BisGMA-based [50]. Besides, the flexural strength of the N’Durance resin composite was reduced significantly by around 73% after 4 months of storage in ethanol [50]. Such results may suggest that dimer acid-based resin composite needs further investigations to optimize its clinical benefits.

### 4.4. Tricyclodecane Urethane-Based Resin Composite

In this class of low shrinkage resin composite, the reaction of hydroxyalkyl (meth) acrylic acid esters with diisocyanates is prepared to insert the urethane group within the methacrylic acid derivatives. The rigidity of tricyclodecane urethane-based resin composite is believed to be the reason why this monomer results in low polymerization shrinkage [75]. Venus Diamond (Venus Diamond, Heraeus Kulzer, Hanau, Germany) is the first commercially available tricyclodecane urethane-based resin composite. Venus Diamond was associated with acceptable mechanical properties after four weeks of storage in water and alcohol compared to other conventional resin composite materials [75]. It was also associated with lower polymerization stress compared to SBRC and dimer acid-based resin composite. However, the volumetric shrinkage was slightly higher than SBRC but still better than the dimer acid-based resin composite [44]. The contraction stress that results from light activation was also investigated. Venus Diamond was found to induce the lowest contraction stress (0.66 ± 0.12 MPa) compared to SBRC (0.92 ± 0.09 MPa) and other MRBC systems [76]. Besides, tricyclodecane urethane was found to achieve an excellent degree of conversion with high biocompatibility [71], indicating the potential of this monomer to be used in future resin composite formulations to minimize the polymerization stress and shrinkage.

### 4.5. Ormocers

Organically modified ceramics (ormocers) are composed of organic and inorganic components. This class of resin composites contains a silicon dioxide network, which is combined with urethane or carboxy-functionalized methacrylate alkoxysilanes using a sol-gel approach [77]. Definite^®^ (Degussa AG, Hanau, Germany) and Admira^®^ (Voco GmbH, Cuxhaven, Germany) are the two commercially available ormocers resin composites. The mechanical properties and shrinkage behavior of ormocers were reported to be comparable to the conventional resin composite [75]. Another investigation reported the use of ormocers synthesized from amine or amide dimethacrylate trialkoxysilanes but the degree of conversion was very low [78].

### 4.6. Thio-Urethane Oligomers

Thio-urethane oligomers have high toughness and a homogenous network, which results in a material with high fracture resistance [79]. The use of thio-urethane oligomers with methacrylate matrices was found effective in reducing the polymerization stress as well as improving the toughness of resin-based materials. Incorporating 10 and 20 wt.% of thio-urethane oligomers into a resin cement containing BisGMA, TEGDMA, and UDMA significantly increased the flexural strength and micro-tensile bond strength compared to the control without affecting the degree of conversion. Moreover, the polymerization stress was reduced by more than 50% when 20 wt.% of thio-urethane oligomers were used [80]. In another investigation, thio-urethanes oligomers were added into a resin composite formulation containing BisGMA and TEGDMA at a 70:30 ratio with a filler load of 70 wt.%. 20% of thio-urethane oligomers reduced the polymerization stress by around 35%. Additionally, the degree of conversion, fracture toughness, flexural strength, and elastic modulus values were significantly improved [81]. Such findings propose that the clinical longevity of resin composites containing thio-urethane oligomers could be improved not only by reducing the polymerization stress but also due to the significant improvement in the polymerization kinetics and mechanical properties.

### 4.7. Thiol-Ene- and Thiol-Norbornene-Based Systems

Thiol-ene-based resins were investigated to replace the use of methacrylate-based resins [82]. Unlike conventional resins, thiol-ene-based resins polymerize mechanism growth includes two steps; first, the addition of a thiyl radical to a functional group, and second, which involves thiyl radical propagation, chain transfer steps, and radical carbon propagation [83].

Thiol-ene polymerization uniquely generates a delayed gel point transformation and contributes to lower volumetric shrinkage, making it very advantageous to use dental resins [84]. Polymerizing the thiol-ene system with methacrylate can reduce the generated stress due to the different patterns concerning the reaction of the two materials and the thiol-ene to act as solvent and chain transfer [85]. A new system called thiol-norbornene was synthesized by integrating norbornene monomers into the thiol-ene systems to improve mechanical properties [86]. The use of ethoxylated bisphenol-A dimethacrylate (EBPADMA) monomer with thiol-ene and thiol-norbornene systems significantly reduced shrinkage stress values compared to resin systems containing BisGMA-TEGDMA or EBPADMA-TEGDMA. The EBPADMA-thiol-ene and EBPADMA-thiol-norbornene resins’ flexural strength values were significantly higher than the EBPADMA-TEGDMA resin and lower than the BisGMA-TEGDMA resin [87].

## 5. Dynamic Covalent Chemistry and Stress Relaxation

The interfacial area in a composite that integrates filler particles of considerably differing modulus in relation to the resin phase performs to concentrate stress and turns out to be a primary factor for composite failure. Since the inorganic fillers occupy a massive surface area in the resin structure, this causes the particle–polymer junction to be fundamentally responsible for a huge amount of chemical and physical phenomena that might affect the required properties [1]. These inorganic fillers can cause stress concentration due to their low thermal expansion and low modulus. This behavior will lead to void formation and composite interfacial debonding, which will influence the failure of composite materials significantly [87,88].

Surface modification of nanoparticles and self-assembled monolayers (SAMs) [89] are techniques that were established to bond the fillers to the resin to advance the connections between the filler particles and the adjoining polymer [89,90]. Yet, these methods progress the efficacy of transferring stress at the interface. However, they do not enable stress relaxation nor eradicate the issue of stress concentration. The technology of using dynamic covalent chemistry (DCC) to produce stress relaxation through triggered interfacial bond exchange whilst preserving a strong, covalent chemical attachment among the filler and the matrix phases is a new adaptive interfacial method that will control the composites nanomaterials limitations [87].

Many studies have implemented the DCC approaches through the resin phase, such as reversible exchange reactions using covalent adaptable networks (CANs). These polymers are cross-linked and are efficient in internal stress relaxation in reaction to the triggering stimulus application such as heat or light through reordering the network’s bonds [91]. Those approaches include addition–fragmentation chain transfer moieties (trithiocarbonate [92,93] or allyl sulfide [94,95]), thiuram disulfides [96], cinnamates [97], metal-catalyzed transesterification [98,99], or reversible addition reactions—for example, the Diels–Alder reaction [100].

Reversible addition–fragmentation chain transfer (RAFT) was studied to synthesize polymer networks in bulk materials, which involves a process of reversible radical-mediated bond exchange with excellent stress relaxation phenomena [101,102,103,104,105]. All of these approaches improve the stress transfer; however, they fail to employ the required DCC where stress is concentrated at the polymer-matrix interface, leading to failure of the composite. Hence, the improvement in the implementation of contemporary resin restorations through relieving the interfacial stresses via DCC methodologies that focuses on the fillers-resin interfacial region without weakening the polymerized material mechanical properties is an important goal. An earlier study developed adaptive interfaces containing RAFT that used silica nanoparticles covalently bonded to a static thiolene resin and facilitated the exchange of bonds solely at the filler-resin interfacial region [87]. As the interfacial area between the inorganic filler particles and the organic resin is recognized as a stress concentrator, isolating the exchange of bonds to happen exclusively at the interface where the stress is condensed allowed relaxation and significantly enhanced the performance of composites. The addition of RAFT into inert thiolene composites has minimized the stresses as well as matches the light-induced radical-mediated methacrylate polymerization. On the other hand, it was found to be altering some properties, such as fracture toughness and Young’s modulus [104]. Another study applied the adaptive interface approach to a contemporary dental resin through the introduction of thiol–thioester (TTE) moieties that stimulate filler–resin bonding and insistent relaxation of stresses at the resin-filler interface without modifying the resin’s formulation [106]. An essential dissimilarity between the TTE, DCC, and RAFT methods is that the RAFT-based methodology is active to facilitate the exchange of bonds only in the presence of radicals in the polymerization process. On the other hand, the TTE method is a nucleophile or base-catalyzed, and therefore, the bond exchange procedure is assumed to continue extended following the initial polymerization is completed in these composites. Hence, it is anticipated that stresses at the interface that occur following polymerization, e.g., because of mismatch in the thermal expansion, would also be capable to relax. Additionally, the TTE method does not necessitate a thiyl radical for the catalyzation of the exchange reaction and is consequently capable of polymerization of the multi-methacrylate resin.

The TTE approach is exhibited to allow relaxation of stress in these greatly stressed, critical interfacial areas. This phenomenon of relaxation continues, even when the DCC is absent in the resin phase due to the catalytic approach, which induces significant improvement in the performance of composites with 30% polymerization stress reduction, 40% flexural strength improvement, and 60% flexural modulus improvement. This method represents an essential clinical value in extending the lifetime of dental composite restorations through interfacial stresses relaxation while enhancing the mechanical properties [104,105].

## 6. Resistance of Dental Resins to Hydrolytic and Enzymatic Degradation in the Oral Environment

Dimethacrylate polymers contain unfavorable ester groups at the end of their chemical chain. Most of the binding ester groups could eventually degrade by esterase enzymes when stimulated intraorally by saliva, cariogenic bacterial products, acidic or basic stimulus, causing hydrolysis and degradation at or adjacent to the tooth-filling interface [9]. Short-term service of these structures combined with unreacted monomers leaching, bisphenol A (BPA), and products degradation from these structures could necessitate numerous dental rework and might elevate other health concerns [106,107].

Current investigations are intended to develop composite resin restorative materials that are greater in properties and durability than the existing polymer restorative materials. Ether-based monomers had shown superior hydrolytic degradation resistance to the currently used ester-containing monomers (esterase) Bis-GMA/TEGDMA in the oral environment [108]. For instance, three ether-based co-polymerizable monomers, erythritol divinylbenzyl ether (E-DVBE), triethyleneglycol divinyl benzyl ether (TEG-DVBE), and the reaction products of vinylbenzylglycidylether with N-tolylglycine salts (NTG-VBGE) were produced, refined, and assessed as alternatives for the presently used dimethacrylate monomers.

The E-DVBE and TEG-DVBE contain two terminal double bonds, which each can copolymerize through the current dental systems of photopolymerization. TEG-DVBE is the diluent monomer used to obtain good handling properties of composite systems and has proven stability against hydrolytic challenges and estrade degradation [39].

Recently, a remarkable low-shrinkage stress resin was developed using urethane dimethacrylate (UDMA) and hydrolytically stable (TEG-DVBE) copolymers. This resin system has unique polymerization kinetics that includes a slower polymerization rate, resulting in a delayed gel point. The longer time taken by the UDMA/TEG-DVBE composite to reach rigidity allowed easier stress relaxation and prevented excessive contraction stress accumulation [109]. In addition, TEG-DVBE can reduce material degradation by improving the resistance to salivary hydrolysis [39,109]. Recent investigations exhibited that 1:1 UDMA:TEG-DVBE molar ratio (U1V1) composites, Figure 3, have good physical and mechanical properties. The tested refractive index, water sorption and solubility, elastic modulus, and polymerization shrinkage are equivalent to the traditional composites Bis-GMA/TEGDMA. They outperformed the controls with respect to flexural strength and fracture toughness, degree of conversion (27.7% increase), and polymerization stresses (57% reduction) compared to the control. Bis-GMA/TEGDMA controls exceeded the experimental composites in the hardness, but they were still above the clinically required threshold of approx. 0.4 GPa [39].

### 6.1. Using Hydrolytically Stable Monomers with Antibacterial and Remineralizing Properties in Composite Resins

While the dilemma of shrinkage stress and secondary caries has long been a concern in the dental community, the need for bioactive low-shrinkage-stress composite is highly desired to improve restoration durability. Several clinical investigations and reviews have demonstrated that secondary caries are a significant reason for posterior composite restoration failure [110]. Secondary caries are defined as primary caries occurring at the restoration-tooth interface due to bacterial biofilm accumulations at the margins [16]. Extensive exposures to sugars and an imbalance in dental biofilm ecology shift the commensal bacteria to a cariogenic condition with increasing production of demineralized acids and subsequent caries formation in the dental hard tissues and increase the demand for the restoration replacement [109,110].

Several investigations have been conducted to fabricate novel low-shrinkage-stress resin with antibacterial and remineralization properties to overcome the shortage in commercially available composites. The researchers developed a hydrolytically stable low-shrinkage-stress polymer containing UDMA and TEG-DVBE (U1V1). It was mixed with dimethylaminododecyl methacrylate monomer (DMAHDM)—with a carbon chain length of 16, which is a highly effective, contact-killing, long-lasting antibacterial material [110,111,112,113,114]. Moreover, adding bioactive nanoparticles of amorphous calcium phosphate (NACP) was advantageous to dental tissues to regenerate the lost minerals and buffer the acidic environment by releasing calcium and phosphate ions to the enamel and dentin under low pH conditions [110,111,112,113,114].

Bhadila et al. showed that U1V1 resin with 3% DMAHDM and 20% NACP did not affect the mechanical properties of the set composite. It also showed calcium and phosphate ions released under an acidic environment to accommodate the demineralization process and enhance hardness in the enamel slaps. Moreover, the polymerization stress of the new composite was 40% lower than that of traditional composite control [111]. Also, the new composite has a strong antibiofilm activity that decreased the S.mutuns colonies to the minimum levels [112,113]. Another study conducted by Filemban et al. concluded that the novel resin composite reduced the stress without compromising other properties, Figure 4. Increasing the DMAHDM content increased the antibacterial effect in a dose-dependent manner [114].

In a study conducted by Albeshir et al., they formulated a novel flowable composite for minimally invasive treatment usages. 5% DMAHDM and 20% NACP were incorporated into 45% U1V1 resin and 30% glass. Paste flowability was within the ISO requirement. Ultimate micro tensile strength and surface roughness were similar to the control, Virtuoso flowable composite (*p* > 0.05). Moreover, the antibacterial response of DMAHDM resin was assessed by using biofilms of the human saliva-derived microcosm model. The biofilm colony-forming unit values were reduced by 5–6 logs (*p* < 0.05). Biofilm metabolic activity was also substantially reduced compared to the control composite (*p* < 0.05). The novel bioactive flowable nanocomposite is promising to be used as pit and fissure sealants and as fillings in conservative cavities to inhibit recurrent caries and increase restoration longevity, as shown in Figure 5 and Figure 6 [110].

## 7. Conclusions

The present review reflects the significant improvements of dental resin to enhance the fillings’ functionality. The findings of this paper revealed the updated information on polymerization shrinkage and its associated stresses and discussed their clinical impacts. In addition, it demonstrated the researchers’ techniques of measuring the shrinkage for better prediction and understanding of the resin composites’ performance. It also highlights the existence of the most effective strategy that could significantly reduce and control polymerization shrinkage stress by developing unique resin formulations to have made a more significant contribution to reducing stress. Finally, it presents the latest investigation of bio-active nanocomposite that will reduce secondary caries, enhance the restoration’s longevity, and maintain the patient’s oral health.

## Figures and Tables

**Figure 1 materials-15-02951-f001:**
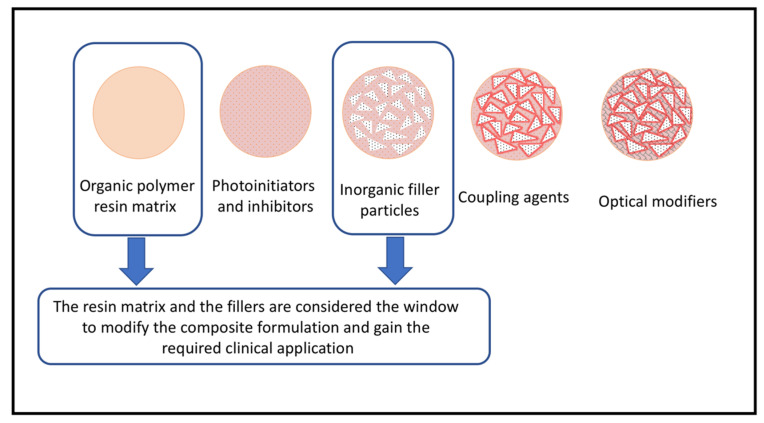
The illustration shows the main components of composite resin restorations.

**Figure 2 materials-15-02951-f002:**
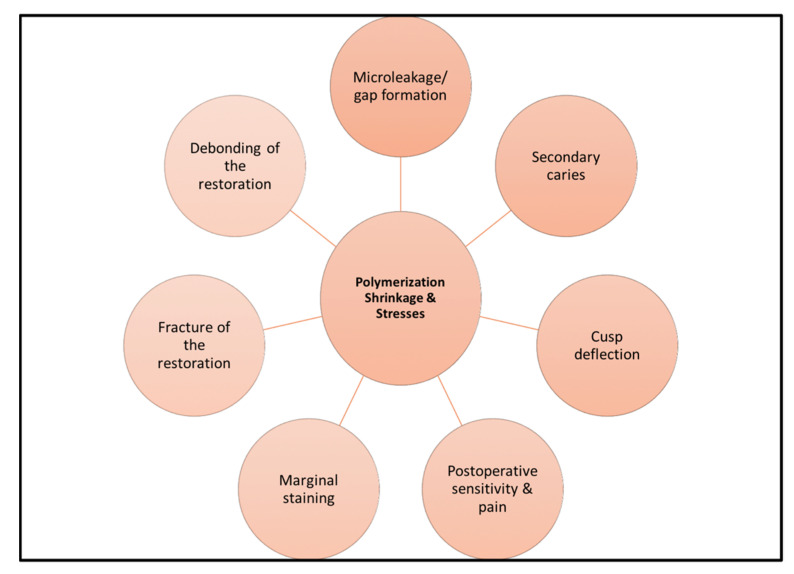
The illustration shows the side effects of polymerization shrinkage and shrinkage-related stresses.

**Figure 3 materials-15-02951-f003:**
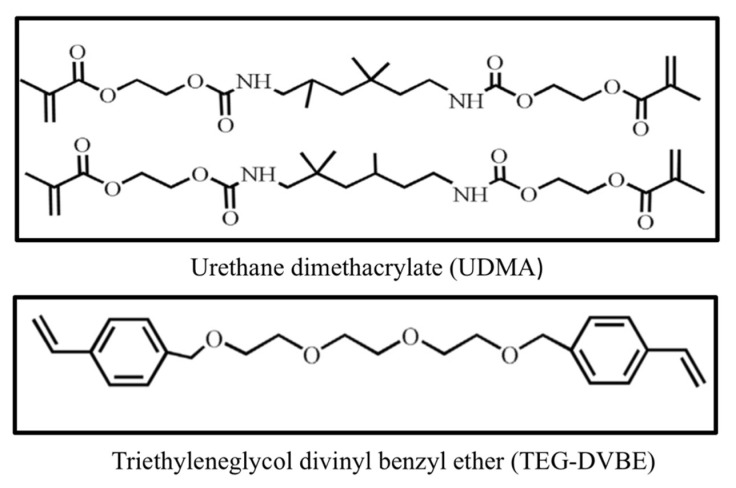
Chemical structure of the resins used in hydrolytically stable, low-shrinkage-stress composites.

**Figure 4 materials-15-02951-f004:**
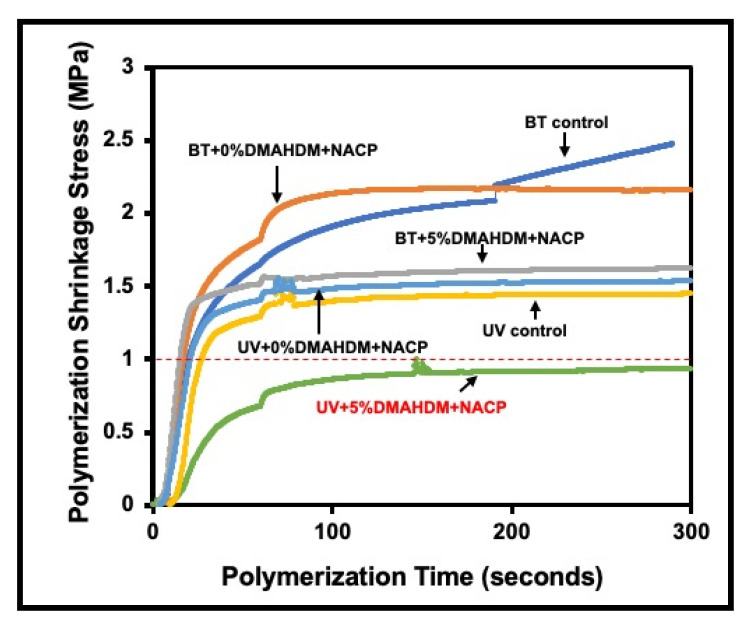
Polymerization shrinkage stress vs. polymerization time (*n* = 3). The low-shrinkage-stress UV resin composite groups showed delayed development of polymerization stress and lower shrinkage stresses (*p* < 0.05) compared to BisGMA + TEGDMA (BT) groups. (Dark blue curve: Experimental control resin composite: 35% BT + 65% glass (denoted as BT control), Orange curve: 35% BT + 0% DMAHDM + 20% NACP + 45% glass (denoted as BT + 0%DMAHDM + NACP), Gray curve: 30% BT + 5% DMAHDM + 20% NACP + 45% glass (denoted as BT + 5% DMAHDM + NACP), Yellow curve: Experimental control resin composite: 35% UV + 65% glass (denoted as UV control), Light blue curve: 35% UV + 0% DMAHDM + 20% NACP + 45% glass (denoted as UV + 0% DMAHDM + NACP), Green curve: 30% UV + 5% DMAHDM + 20% NACP + 45% glass (denoted as UV + 5% DMAHDM + NACP)), Adapted with permission from reference [114]. Copyright 2021 Elsevier.

**Figure 5 materials-15-02951-f005:**
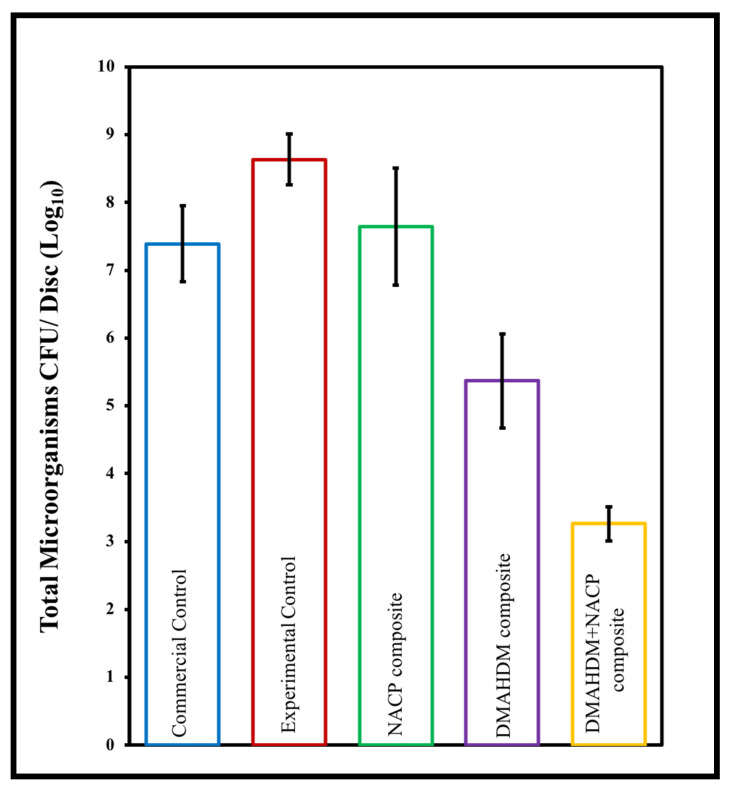
Mutans streptococci of 2-day biofilm colony forming units CFU. Dissimilar letters indicate values that are significantly different from each other (*p* < 0.05). Notice that groups with antibacterial agent DMAHDM have the more bacterial log reduction. Commercial Control composite (High-viscosity Sealant/Flowable Composite Virtuoso, DenMat, Lompoc, CA, USA). 50% UV + 50% glass (Experimental Control composite), 45% UV + 20% NACP + 35% glass (UV + NACP composite), 45% UV + 5% DMAHDM + 50% glass (UV+DMAHDM composite), 45% UV + 5% DMAHDM + 20% NACP + 30% glass (UV + DMAHDM + NACP + 30glass), * UV indicates low-shrinkage-stress resin composed of UDMA + TEG-DVBE, Adapted with permission from reference [110]. Copyright 2021 Elsevier.

**Figure 6 materials-15-02951-f006:**
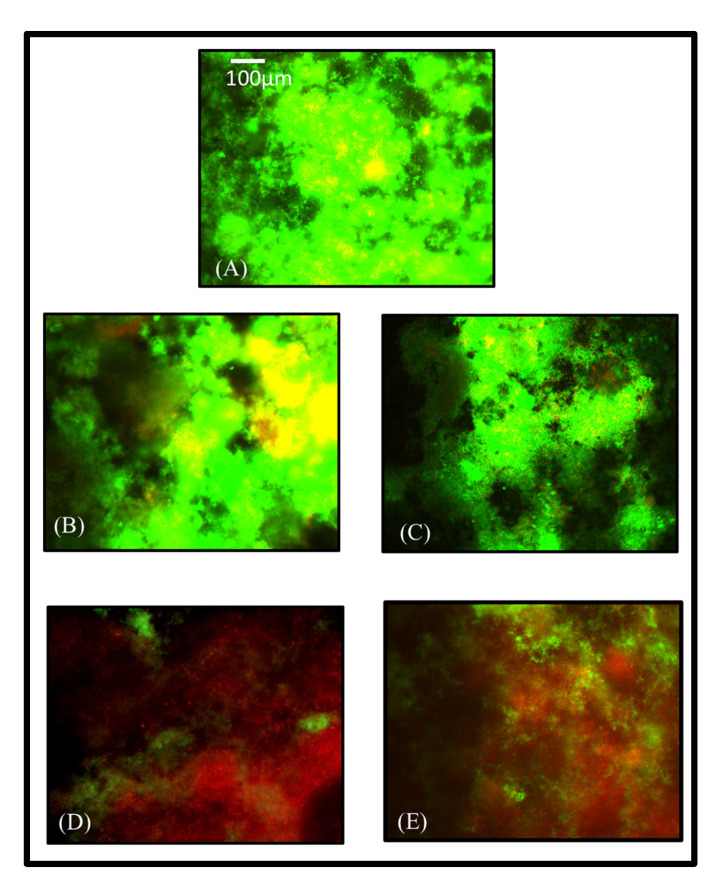
Representative live/dead staining images of biofilms: (**A**) commercial control, (**B**) experimental control, (**C**) NACP group, had no antibacterial agent, and the bacteria was alive (in green). (**D**,**E**) had 5% DMHADM, showed dead bacteria (in red). Commercial Control composite (High-viscosity Sealant/Flowable Composite Virtuoso, DenMat, Lompoc, CA, USA), 50% UV + 50% glass (Experimental Control composite), 45% UV + 20% NACP + 35% glass (UV + NACP composite), 45% UV + 5% DMAHDM + 50% glass (UV + DMAHDM composite), 45% UV + 5% DMAHDM + 20% NACP + 30% glass (UV + DMAHDM + NACP + 30glass), * UV indicates low-shrinkage-stress resin composed of UDMA + TEG-DVBE. Adapted with permission from reference [110]. Copyright 2021 Elsevier.

## Data Availability

Not applicable.

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
