# Peer review of "Low-Shrinkage Resin Matrices in Restorative Dentistry-Narrative Review"

_materials, 2022, doi:10.3390/ma15082951_

Round 1
Reviewer 1 Report
Dear authors,
I reviewed this paper with interest.
There are several issues that should be addressed in the manuscript before further consideration for publication.
Page 2.
Composites are a three-dimensional combination of four major chemically different components, Figure 1.
Where is Figure 1.
Page 3. Figure1.
Please reconsider figure 1.
The side effect of polymerization shrinkage and shrinkage-related stresses are mentioned in page 3.
However, Microleacage, gap formation and marginal staining are caused by low bonding performance between adhesive material and tooth.
Enamel cracks might be caused by diamond bar at tooth cavity preparation, material shrinkage or low bonding performance.
The cause is still unknown.
I think better “Enamel cracks” is removed form Figure 1.
And you should discuss about below side effects.
Page 9 to 10.
Replace 3.4 with 3.5, 3.6, 3.7.
References,
The style of writing references are not the same as the journal style.
Author Response
Reviewer #1:
Dear authors,
I reviewed this paper with interest. There are several issues that should be addressed in the manuscript before further consideration for publication.
Thank you for your comments. We appreciate your time reviewing our manuscript. Below is our detailed response to each comment.
Page 2. Composites are a three-dimensional combination of four major chemically different components, Figure 1.
Where is Figure 1.
Page 3. Figure1.
Please reconsider figure 1.
Thanks for the valuable comment. We have corrected the citation of figure 1 and the picture in the text to be of better quality. Figure 1 is placed on page #3.
The side effect of polymerization shrinkage and shrinkage-related stresses are mentioned in page 3.
However, Microleakage, gap formation and marginal staining are caused by low bonding performance between adhesive material and tooth.
Enamel cracks might be caused by diamond bar at tooth cavity preparation, material shrinkage or low bonding performance.
The cause is still unknown.
I think better “Enamel cracks” is removed form Figure 2.
And you should discuss about below side effects.
Your comments are positively precise. The causes of enamel cracks are various and overlapped. “Enamel cracks” had been removed from Figure 2. Figure 2 is placed on page #4.
Page 9 to 10.
Replace 3.4 with 3.5, 3.6, 3.7.
Done. Marked in red.
References,
The style of writing references is not the same as the journal style.
The references have been updated and marked with red.
"Please see the attachment"

Reviewer 2 Report
Abstract: Well written.
Introduction: break the introduction in subchapters. It is quite long. It will offer a better reading for researchers who are interested in your topic.
Figure 1,2 and 6. Insert a better quality.
5. Resistance of Dental Resins to Hydrolytic and Enzymatic Degradation in the Oral Environment: It is really great that you discussed this topic. I would strongly suggest you to introduce some paragraphs about the influence of dental resin also in patients with systemic disease. Here are some sources of inspiration (10.3390/polym14030569 ; 10.3390/biomedicines9111618).
Conclusion: Write what are the great improvements of dental resin.
Please revise the English language. There are some typos and language mistakes.
Overall, you did a really amazing review. I suggest only minor rev.
Good luck
Reviewer
Author Response
Comments and Suggestions for Authors
Abstract: Well written.
Thank you for your comments. We appreciate your time reviewing our manuscript.
Introduction: break the introduction in subchapters. It is quite long. It will offer a better reading for researchers who are interested in your topic.
The long sentences have been broken down into shorter and clearer ones. Marked on red font.
Figure 1,2 and 6. Insert a better quality.
The figures have been replaced with better quality ones.
Figure 1 placed on page #3
Figure 2 placed on page # 4
Figure 6 placed on page # 16
- Resistance of Dental Resins to Hydrolytic and Enzymatic Degradation in the Oral Environment: It is really great that you discussed this topic. I would strongly suggest you to introduce some paragraphs about the influence of dental resin also in patients with systemic disease. Here are some sources of inspiration (10.3390/polym14030569 ; 10.3390/biomedicines9111618).
This comment is significant. Hydrolytic and Enzymatic Degradation in the Oral Environment is a very valuable topic because it dramatically affects the longevity of the restoration as well as has systematic effects. We will highly consider this important topic for future papers. Thank you.
Conclusion: Write what are the great improvements of dental resin.
The conclusion has been updated. “The present review reflects the significant improvements of dental resin to enhance the fillings' functionality. The findings of this paper revealed the updated information of polymerization shrinkage and its associated stresses and discussed their clinical impacts. Also, it demonstrated the researchers' techniques of measuring the shrinkage for better prediction and understanding of materials' performance. It also highlights the existence of the most effective strategy that could significantly reduce and control polymerization shrinkage stress by developing unique resin formulations to have made a more significant contribution to reducing stress. Finally, it presents the latest investigation of bio-active nanocomposite that will reduce secondary caries, enhance the restoration's longevity, and maintain the patient's oral health.”
Please revise the English language. There are some typos and language mistakes.
Language and typo mistakes have been checked.
Overall, you did a really amazing review. I suggest only minor rev.
So grateful for your great support.

Reviewer 3 Report
Dear Authors
This review focuses on the methods for testing the shrinkage, as well as formulations of resinous matrices available to reduce polymerization shrinkage and its associated stress. Although the topic is interesting, the Materials and Methods section lack, and it is detrimental for the scientific meaning of the Ms. I suggest describing the decision making in choosing the articles of this review. Moreover, the title should be rephrased and modified based on the review type: Narrative Review, Systematic Review, etc.
Then, for these and other reasons I suggest a Major revision before the publication.
Abstract
Specify the meaning of “transformational”
Composite, resin composite or polymeric materials have the same meaning? The authors should use only one expression in the abstract and along the text
Following the abstract, this Ms is a review, and it should be specified in the title.
Main Text
Line 3 page 2: Please rephrase “Moreover, choosing the best material to replace missing tissue from disease or trauma”
Line 14 page 2: Authors said, “patients are more attracted to tooth-colored restorations” and “...place artistic restoration ”. The final goal of resin composite is based on their capability to be mimetic with the surrounded dental tissue.
Line 20 page 2: “This bondability - achieved by applying an adhesive- seals the margins and reinforces the remaining tooth structure by forming micromechanical interaction between these restorations and etched enamel or dentin” The Authors should consider and discuss the difference between the vestibular and lingual surface of permanent and deciduous teeth, since it might influence the bondability of resin materials.
Line 26 page 2: Figure 1 is not related to the of four major chemically different components. The Authors should better describe this aspect.
Line 4 page 4: In this paragraph the Authors should provide references of their sentences.
Line 25 page 4: “light-curing protocols”, The Authors should provide references, for example the step luting protocol or the stepwise.
The section “1. Approaches for testing resin composite shrinkage” the number of this paragraph is the same of introduction. Please correct.
Page 5: The formula of Linershrinkage should be better written
Line 30 page 6: μCT was used also for the assessments of margins and internal fit of class II cavities. Please discuss and add references
Paragraph 2.6: Authors said “… linear vertical displacement transducer (LVDT)”. The Authors used this abbreviation, however LVDT was not used any further. I suggest removing such abbreviation. The same consideration along the text.
Page 13: Figure 3. Specify the abbreviation in the figures, UDMA and TEG-DVBE.
line 1 page 14: Correct the style of “Bhadila et al.”
Figure 5: The style of Figure Legend should be corrected following the author`s guidelines of the Manuscript.
Line 3 page 17: “Finally, it presents the latest investigation of bioactive nano-composite that will reduce secondary caries, enhance the restoration's longevity, and maintain the patient's oral health.” In this manuscript only few references regarded bioactive nanocomposite, then this conclusion should be rephrased.
Author Response
Dear Authors
This review focuses on the methods for testing the shrinkage, as well as formulations of resinous matrices available to reduce polymerization shrinkage and its associated stress. Although the topic is interesting, the Materials and Methods section lack, and it is detrimental for the scientific meaning of the Ms. I suggest describing the decision making in choosing the articles of this review. Moreover, the title should be rephrased and modified based on the review type: Narrative Review, Systematic Review, etc.
Thank you for your comments. We appreciate your time reviewing our manuscript. Below is our detailed response to each comment.
I suggest describing the decision making in choosing the articles of this review. Moreover, the title should be rephrased and modified based on the review type: Narrative Review, Systematic Review, etc.
Thank you. This paper is Narrative review. The little has been updated.
Then, for these and other reasons I suggest a Major revision before the publication.
Sure. Following the valuable reviewers’ comments, a major change has been processed.
Abstract
Specify the meaning of “transformational”
the word “transformational” has been removed.
Composite, resin composite or polymeric materials have the same meaning? The authors should use only one expression in the abstract and along the text
Thank you. resin composite has been used for all text.
Following the abstract, this Ms is a review, and it should be specified in the title.
the title has been changed to (Low-Shrinkage Resin Matrices in Restorative Dentistry. Review)
Main Text
Line 3 page 2: Please rephrase “Moreover, choosing the best material to replace missing tissue from disease or trauma”
This sentence has been removed.
Line 14 page 2: Authors said, “patients are more attracted to tooth-colored restorations” and “...place artistic restoration ”. The final goal of resin composite is based on their capability to be mimetic with the surrounded dental tissue.
You are absolutely right...
Line 20 page 2: “This bondability - achieved by applying an adhesive- seals the margins and reinforces the remaining tooth structure by forming micromechanical interaction between these restorations and etched enamel or dentin” The Authors should consider and discuss the difference between the vestibular and lingual surface of permanent and deciduous teeth, since it might influence the bondability of resin materials.
This comment is significant. The difference between in the bonding of the vestibular and lingual surface of permanent and deciduous teeth, which affects bondability of resin materials. We will highly consider this important topic for future papers. Thank you.
Line 26 page 2: Figure 1 is not related to the of four major chemically different components. The Authors should better describe this aspect.
The main four components of resin composites are: polymer resin matrix, inorganic filler, coupling agents and optical modifiers. The figure also shows that the resin and the filler are the window for modification and addition of other substances to change and improve the net composite restorations.
The descriptions have been added to the text.
Line 4 page 4: In this paragraph the Authors should provide references of their sentences.
The references have been added.
Line 25 page 4: “light-curing protocols”, The Authors should provide references, for example the step luting protocol or the stepwise.
The references have been added.
The section “1. Approaches for testing resin composite shrinkage” the number of this paragraph is the same of introduction. Please correct.
Thank you. The numbers have been changed and checked.
Page 5: The formula of (Liner shrinkage) should be better written
Liner shrinkage formula has been updated.
Line 30 page 6: μCT was used also for the assessments of margins and internal fit of class II cavities. Please discuss and add references
The reference has been added.
μCT is a handy tool for measuring many aspects of dentistry. This part focuses on measuring the polymerization shrinkage and its associated stress. We will highly consider this important topic for future papers. Thank you.
μCT is very useful tool for mearing many
Paragraph 2.6: Authors said “… linear vertical displacement transducer (LVDT)”. The Authors used this abbreviation, however LVDT was not used any further. I suggest removing such abbreviation. The same consideration along the text.
The (LVDT) has been removed from the paragraph.
Page 13: Figure 3. Specify the abbreviation in the figures, UDMA and TEG-DVBE.
The abbreviation has been added to the figure #3, page #13
line 1 page 14: Correct the style of “Bhadila et al.”
“Bhadila et al”. has been changed.
Figure 5: The style of Figure Legend should be corrected following the author`s guidelines of the Manuscript.
The Figures legends have been corrected.
Line 3 page 17: “Finally, it presents the latest investigation of bioactive nano-composite that will reduce secondary caries, enhance the restoration's longevity, and maintain the patient's oral health.” In this manuscript only few references regarded bioactive nanocomposite, then this conclusion should be rephrased.
Yes, you are right. There are just few references regarding this novel topic “Hydrolytically stable, Low shrinkage stress resin with antibacterial and remineralization properties”. This part makes our review pretty special and novel because no other reviews spots the light on this aspect.
Reviewer 4 Report
Dear authors, I have seldom read such a well done "Review" article.
This paper will allow experienced and less experienced dentists to have a scientifically sound guide to understanding different composites. It is also an article where all the main composites can be listed, including those of the latest generation, which is an added value. English is of a high standard.
Congratulations!!! I just have a suggestion: please add reference after this statement :Besides their excellent esthetic properties, resin composites have low thermal conductivity, radiopacity, no mercury, and no galvanic currents
Author Response
We are so grateful for your unique comments. Your words are so supportive and engorgement. We appreciate your time reviewing our manuscript. The reference to that sentence has been added, page #2, paragraph #2, marked in red font.

Round 2
Reviewer 3 Report
Dear Authors
Most of my comments were responded correctly. However, I strongly recommend Authors to provide Material and Methods section. Moreover, although a narrative review should summarize different aspect of a specific topic, the following comments might increase the scientific sound of the Ms.
Page 2: “The bonding succeeded by using acids that etched the dental tissues to form numerous micromechanical interactions between these restorations and enamel orand dentin [8,9].”. Since the new outcomes regarding the difference in composition between vestibular and lingual enamel and dentin I suggest discussing this topic.
Page 4: “light-curing protocols”. The Authors should discuss and provide references regarding other curing protocols, for example the step luting protocol or the stepwise.
Page 6: The Authors did not discuss the use of μCT was used also for the assessments of margins and internal fit of class II cavities. At least few studies about this topic, with reference.
Although, in the last my comment the Authors said “Yes, you are right. There are just few references regarding this novel topic “Hydrolytically stable, Low shrinkage stress resin with antibacterial and remineralization properties”. This part makes our review pretty special and novel because no other reviews spots the light on this aspect.”. They should provide at least the methods used for the literature research.
In the conclusion: “ Also, it demonstrated the researchers' techniques of measuring the shrinkage for better prediction and understanding of materials' performance”. Please rephrase specifying the shrinkage of which material.
Author Response
Reviewer #3: (Round 2) Also, please see the attachment
Most of my comments were responded correctly. However, I strongly recommend Authors to provide Material and Methods section.
Yes, materials and methods are indeed important. Please note that our Section 2, Approaches for testing resin composite shrinkage, contains “the methods”. Our Section 3, Resin formulations attempt to reduce shrinkages and their stress, contains “the materials”. Alternatively, we could take out the materials and methods from these two sections and form a separate new section, but we found that such an arrangement makes the paper messy. We value you comment that materials and methods are important, and please note that they are contained in Sections 2 and 3.
Moreover, although a narrative review should summarize different aspect of a specific topic, the following comments might increase the scientific sound of the Ms.
Yes, please see point-by-point response below.
Page 2: “The bonding succeeded by using acids that etched the dental tissues to form numerous micromechanical interactions between these restorations and enamel or and dentin [8,9].”. Since the new outcomes regarding the difference in composition between vestibular and lingual enamel and dentin I suggest discussing this topic.
Thanks for this significant comment. The factors that affect adhesion are many, and this is a broad topic to discuss. Since we are focusing on the composite as a material, we wrote more about the composition and the features. Moreover, the following paragraph has been added, page # 2, paragraph #2, line #17.
- "This bondability - achieved by applying an adhesive - seals the margins and reinforces the remaining tooth structures. The bonding succeeded by using acids that etched the dental tissues to form numerous micromechanical interactions between the restoration and enamel and dentin [8]. The variety of dental tissues also plays a significant role in the adhesion process, for example, enamel vs. dentin tissues, permanent vs. deciduous teeth, or vestibular vs. lingual dental tissues [9]. Because of the material's bonding, it has increased application in modern preventive and conservative dentistry."
- Also, reference #9 has been updated.
Page 4: “light-curing protocols”. The Authors should discuss and provide references regarding other curing protocols, for example the step luting protocol or the stepwise.
Thanks for this note. The following updates have been added, page #4 paragraph #3.
“In addition, polymerization shrinkage stresses can be reduced by applying techniques such as incremental layering, a stress-absorbing base, or a liner [6,12]. Furthermore, decreasing the rate of shrinkage of the composite can be affected by the intensity and mode of curing light, such as soft-start polymerization [12].”
Page 6: The Authors did not discuss the use of μCT was used also for the assessments of margins and internal fit of class II cavities. At least few studies about this topic, with reference.
The part of X‐ray microcomputed tomography has been changed, page #6-7.
“X‐ray microcomputed tomography (μCT) can acquire three-dimensional (3D) structures of the internal content of small objects with high spatial resolution [27]. Biomedical research has been widely accepted for examining bone and tooth structures, visualizing structural features in tissue engineering scaffolds, and assessing the mineral concentration of teeth [28]. Recently, μCT has been used to evaluate the junction of the tooth-adhesive and the marginal adaptation, which correlated with shrinkage strain of the resin composite restorations. The previous study used μCt to examine the dimensional changes of dental resin composites before and after polymerization [28]. The acquired results agreed with the degree of shrinkage achieved by density measurements. μCT gives matching accuracy for different physical states and shapes. Furthermore, μCT results are not affected with air bubbles because they are not included in calculating the dimension of the resin restoration [28].”
Although, in the last my comment the Authors said “Yes, you are right. There are just few references regarding this novel topic “Hydrolytically stable, Low shrinkage stress resin with antibacterial and remineralization properties”. This part makes our review pretty special and novel because no other reviews spots the light on this aspect.”. They should provide at least the methods used for the literature research.
A research library supported the databases search for subject terms, keywords, and text words, related to studies that evaluated the Low-Shrinkage Resin Matrices in Restorative Dentistry. The related articles were searched using Medline (OVID) and EMBASE databases. The investigation method conducted for MEDLINE was followed for EMBASE and appropriately revised to account for vocabulary differences. Searched terms were related to the Low-shrinkage resin matrices used in dentistry and involved but were not limited to antibacterial or antibiofilm, remineralization, resin or composite or nanocomposite, and low shrinkage stress. The searches were limited to peer-reviewed journals. Searches were also limited to English language articles from 2000-to 2022. At the end of the search, only six studies were found.
In the conclusion: “ Also, it demonstrated the researchers' techniques of measuring the shrinkage for better prediction and understanding of materials' performance”. Please rephrase specifying the shrinkage of which material.
You are right. This sentence needs to clarify. Conclusion, Page #17.
In addition, it demonstrated the researchers' techniques of measuring the shrinkage for better prediction and understanding of the resin composites performance.

Round 3
Reviewer 3 Report
Dear Author
For next time, to increase readability of the revised Ms, I suggest avoiding word track changes and highlighting in red the changes. Moreover, the format and style of the Journal should be maintained in the revised version (pages 5-6) and the layout should be improved (page 13, the paragraph name might be put in page 14)
Insert Narrative review in the title.
Materials and Methods: Although the Authors might be disappointed, I suggest putting Materials and Methods section in order to increase the scientific meaning of the work. In the last response, the Authors wrote “A research library supported the databases search for subject terms, keywords, and text words, related to studies that evaluated the Low-Shrinkage Resin Matrices in Restorative Dentistry. The related articles were searched using Medline (OVID) and EMBASE databases. The investigation method conducted for MEDLINE was followed for EMBASE and appropriately revised to account for vocabulary differences. Searched terms were related to the Low-shrinkage resin matrices used in dentistry and involved but were not limited to antibacterial or antibiofilm, remineralization, resin or composite or nanocomposite, and low shrinkage stress. The searches were limited to peer-reviewed journals. Searches were also limited to English language articles from 2000-to 2022. At the end of the search, only six studies were found.”. The Authors should add the selected keywords and the total number of articles, if available directly in the Materials and Methods section. Moreover, Authors might consider other narrative review of the same Journal.
Page 5: Although the Authors change the paragraph “In addition, polymerization shrinkage stresses can be reduced by applying techniques such as incremental layering, a stress-absorbing base, or a liner [6,12]. Furthermore, decreasing the rate of shrinkage of the composite can be affected by the intensity and mode of curing light, such as soft-start polymerization [12].”, they did not provide or even list other different curing modes, adding at least other references. Since “the manifestation of the polymerization shrinkage and the associated stresses are the major problems facing manufacturers and clinician”, a list of different curing mode might be useful for the reader.
Although the part of X‐ray microcomputed tomography has been changed, page #6-7, the Authors used only 1 or 2 references in this paragraph. Please provide other references. For example, about margins and internal fit of different types of cavity.
Author Response
Reviewer #3: (Round 3)
Dear Author
For next time, to increase readability of the revised Ms, I suggest avoiding word track changes and highlighting in red the changes.
Thank you for your comments. The MDPI editors mentioned in the email that “Any revisions made to the manuscript should be marked up using the “Track Changes” function if you are using MS Word/LaTeX, such that changes can be easily viewed by the editors and reviewers.” The new version on MS has no word tracking and no red font highlights.
Moreover, the format and style of the Journal should be maintained in the revised version (pages 5-6) and the layout should be improved (page 13, the paragraph name might be put in page 14).
- The format and style of pages #4-6 have been checked.
- Page#13: paragraph title has been moved to page #14.
Insert Narrative review in the title.
The title has been changed to “Low-Shrinkage Resin Matrices in Restorative Dentistry. Narrative Review”.
Materials and Methods: Although the Authors might be disappointed, I suggest putting Materials and Methods section in order to increase the scientific meaning of the work. In the last response, the Authors wrote “A research library supported the databases search for subject terms, keywords, and text words, related to studies that evaluated the Low-Shrinkage Resin Matrices in Restorative Dentistry. The related articles were searched using Medline (OVID) and EMBASE databases. The investigation method conducted for MEDLINE was followed for EMBASE and appropriately revised to account for vocabulary differences. Searched terms were related to the Low-shrinkage resin matrices used in dentistry and involved but were not limited to antibacterial or antibiofilm, remineralization, resin or composite or nanocomposite, and low shrinkage stress. The searches were limited to peer-reviewed journals. Searches were also limited to English language articles from 2000-to 2022. At the end of the search, only six studies were found.”. The Authors should add the selected keywords and the total number of articles, if available directly in the Materials and Methods section. Moreover, Authors might consider other narrative review of the same Journal.
Materials and Methods section has been added page# 5
- Materials and Methods
This review paper focuses on polymerization shrinkage and stresses in dental composite restorations. It sheds light on the different approaches of physical testing of the stresses and measuring the shrinkage. Furthermore, this article discusses the formulations of resinous matrices available to reduce shrinkage and the associated stresses. In addition, this review updates the reader with cutting-edge research on bioactive low-shrinkage-stress nanocomposites.
For paragraph (6.1), a research library supported the databases search for subject terms, keywords, and text words, that were related to studies on low-shrinkage resin matrices in restorative dentistry. The related articles were searched using Medline (OVID) and EMBASE databases. The investigation method conducted for MEDLINE was followed for EMBASE and appropriately revised to account for vocabulary differences. Searched terms were related to the low-shrinkage resin matrices used in dentistry and involved, but were not limited to, antibacterial or antibiofilm, remineralization, resin or composite or nanocomposite, and low shrinkage stresses. The searches were limited to peer-reviewed journals. Searches were also limited to English language articles from 2000 to 2022. At the end of the search, only six qualified studies were found.
Page 5: Although the Authors change the paragraph “In addition, polymerization shrinkage stresses can be reduced by applying techniques such as incremental layering, a stress-absorbing base, or a liner [6,12]. Furthermore, decreasing the rate of shrinkage of the composite can be affected by the intensity and mode of curing light, such as soft-start polymerization [12].”, they did not provide or even list other different curing modes, adding at least other references. Since “the manifestation of the polymerization shrinkage and the associated stresses are the major problems facing manufacturers and clinician”, a list of different curing mode might be useful for the reader.
The following paragraph and more references have been added page #4,5.
“Furthermore, modifying light-activation protocols, such as soft-start and pulse delay, have been advocated to reduce shrinkage stress [17]. The hypothesis is that starting the polymerization with low intensity produces a reduced amount of free-radicals with a slower polymerization reaction, delaying the vitrification point of composite [17,20]. Thus, it allows more relief of shrinkage stress by prolonging the period that composite can flow [21]. Based on this concept, many light-curing units offer alternative regimens to emit light on pulsatile, ramp, or soft-start modes [20-22]. However, many studies showed that the set composite using modified light-activation protocols might be more prone to degradation, with lower elastic modulus, and higher risk of failure under occlusal loading because of lower strength properties [22,23].
Although the part of X‐ray microcomputed tomography has been changed, page #6-7, the Authors used only 1 or 2 references in this paragraph. Please provide other references. For example, about margins and internal fit of different types of cavity.
References # 33,34 have been added to X‐ray microcomputed tomography part.
(X-ray microcomputed tomography (μCT) can acquire three-dimensional (3D) structures of the internal content of small objects with high spatial resolution [31]. Biomedical research has been widely accepted for examining bone and tooth structures, visualizing structural features in tissue engineering scaffolds, and assessing the mineral concentration of teeth [32]. Recently, μCT has been used to evaluate the junction of the tooth-adhesive and the marginal adaptation, which correlated with shrinkage strain of the resin composite restorations [33,34]. The previous study used μCT to examine the dimensional changes of dental resin composites before and after polymerization [32]. The acquired results agreed with the degree of shrinkage achieved by density measurements. The μCT method gives matching accuracy for different physical states and shapes. Furthermore, the μCT results are not affected with air bubbles because they are not included in calculating the dimension of the resin restoration [32,34].)
